# Environmental Restrictions: A New Concept Governing HIV-1 Spread Emerging from Integrated Experimental-Computational Analysis of Tissue-Like 3D Cultures

**DOI:** 10.3390/cells9051112

**Published:** 2020-04-30

**Authors:** Samy Sid Ahmed, Nils Bundgaard, Frederik Graw, Oliver T. Fackler

**Affiliations:** 1Department of Infectious Diseases, Integrative Virology, University Hospital Heidelberg, 69120 Heidelberg, Germany; Samy.SidAhmed@med.uni-heidelberg.de; 2BioQuant – Center for Quantitative Biology, Heidelberg University, 69120 Heidelberg, Germany; nils.bundgaard@bioquant.uni-heidelberg.de

**Keywords:** HIV-1 spread, cell-free infection, cell–cell transmission, 3D cultures, mathematical modeling, environmental restriction

## Abstract

HIV-1 can use cell-free and cell-associated transmission modes to infect new target cells, but how the virus spreads in the infected host remains to be determined. We recently established 3D collagen cultures to study HIV-1 spread in tissue-like environments and applied iterative cycles of experimentation and computation to develop a first in silico model to describe the dynamics of HIV-1 spread in complex tissue. These analyses (i) revealed that 3D collagen environments restrict cell-free HIV-1 infection but promote cell-associated virus transmission and (ii) defined that cell densities in tissue dictate the efficacy of these transmission modes for virus spread. In this review, we discuss, in the context of the current literature, the implications of this study for our understanding of HIV-1 spread in vivo, which aspects of in vivo physiology this integrated experimental–computational analysis takes into account, and how it can be further improved experimentally and in silico.

## 1. Introduction

As obligate intracellular parasites, the replication of viruses depends on the infection of host cells that support the viral life cycle and the production of viral progeny. In order to establish virus replication in a new host, the virus has to efficiently spread following the initial infection at the portal of entry. The production of infectious progeny and infection of new target cells represents the central mechanism for virus spread. In principle, this can be achieved by the release of virus particles into the extracellular space, which can encounter and infect new target cells (cell-free infection) (Figure 1a). In addition, viruses can be transferred from infected donor cells to uninfected target cells via close physical contact between the cells (cell–cell transmission) (Figure 1b–d). Cell-associated modes of virus transmission include the short-distance transmission of cell-free virus at cell–cell contacts (Figure 1d), the transport of virus particles along or within cell protrusions connecting donor and target cells (Figure 1b,c), as well as cell–cell fusion [1,2], and are generally considered more efficient than cell-free infections. While cell-associated modes of virus transmission have been less explored than cell-free infection, evidence for the use of this transmission mode is steadily increasing and has been documented, e.g., for Vaccinia virus [3], Hepatitis C virus [4], Herpes Simplex virus [5], Epstein–Barr Virus [6] Dengue Virus [7], and the pathogenic human retroviruses Human Immunodeficiency Virus type 1 (HIV-1) and Human T-cell Lymphotropic Virus type 1 (HTLV-1) [8,9,10,11,12]. Most of these viruses are known to be able to spread by cell-free and cell-associated modes of transmission, but some viruses, such as HTLV-I, specialize in cell–cell transmission and appear to exclusively rely on this transfer mode, as cell-free infectious virus can seldom be isolated [12]. For viruses using both cell-free and cell-associated transmission, the relative contribution of each transmission mode to overall spread is difficult to assess, and the pathophysiological relevance of cell-associated transmission often remains unclear. It is therefore not surprising that traditional concepts in virology have focused on cell-free infections, which is still reflected in the majority of experimental studies conducted. 

HIV-1 is an example of a virus for which the modes of transmission are particularly well studied. Initially assumed to spread exclusively via cell-free virus, early studies indicated that infected cells are a much better inoculum to drive virus spread in a new culture than cell-free virus [13]. The demonstration that constant agitation of infected CD4^+^ T cells or physical separation of infected from uninfected cells by transwells disrupts the formation of cell–cell contacts as well as efficient virus spread then suggested that, in fact, cell-associated modes of transmission are essential for efficient HIV-1 spread in CD4^+^ T-cell cultures [14,15]. A large series of imaging-based studies has now established that in addition to infection with cell-free virions, HIV-1 can efficiently spread via cell-cell contacts. Although probably relying on a slightly divergent mechanism, HIV-1 cell–cell transmission is observed between CD4^+^ T cells, for the transfer from dendritic cells to CD4^+^ T cells, between CD4^+^ T cells and macrophages, and between myeloid cells [16]. The cell–cell contacts involved are referred to as virological synapses (VSs) between productively infected donor and target cells, or as infectious synapses when donor cells such as dendritic cells store virus for transmission without being productively infected [17,18,19,20,21,22]. This contact-dependent transmission mode has been found to be much more efficient than cell-free virus uptake, with an estimated 10-fold to 18,000-fold higher efficiency in mediating viral spread [14,23,24,25]. Despite this overwhelming evidence that HIV-1 efficiently uses cell-associated modes of transmission in experimental HIV-1 infection, the technical barriers to studying this aspect of HIV-1 biology in vivo limit the generation of evidence for the use of this transmission mode in the infected host. Only a few studies have reported visualization of cell–cell contacts reminiscent of a VS in vivo [26,27,28], or established the motility of cells loaded with infectious HIV-1 as essential for efficient HIV-1 spread in infected humanized mice [27,29]. By which transmission mode HIV-1 spreads in vivo, and how this might depend on the specific tissue environment, thus remains to be established.

## 2. 3D Culture Systems

Questions such as the relevance of individual virus transmission modes for viral spread and disease progression would be best addressed in in vivo infection models, as these reflect the complex tissue organization and pathogen–host interactions of infected patients [33]. In the case of HIV-1 infection, various humanized mouse models that mirror a range of but not all pathological events of AIDS are available for such studies [34]. However, experimental parameters are very difficult to control in these complex in vivo systems, and the number of experiments that can be conducted is limited by logistic, financial, and/or ethical concerns. In turn, simple monotypic, two-dimensional (2D) cultures do not reflect the complex architecture and cell heterogeneity of HIV-1 target tissue and, even for suspension cells, such as CD4^+^ T cells, quickly organize into a cell monolayer with dense cell packing. In vivo, the physical distance between donor and target cells often is the main barrier for efficient virus transmission, be it via diffusion of cell-free particles or the motility of the cell towards forming cell–cell contacts for virus transmission. Thus, densely packed 2D cultures are not suitable to study such processes. 

As these constraints apply to all areas of biology, major efforts are being undertaken to establish and exploit experimental 3D systems that faithfully reflect specific aspects of tissue organization and physiology. One major area of development involves establishing organ or organoid cultures, either by culturing original tissues obtained from surgery or by reconstituting organ-like tissue ex vivo by differentiation from induced pluripotent stem cells [35,36]. In the case of HIV-1, such organotypic cell systems include cultures of human tonsil tissue as surrogates for lymph node tissues or mucosa explants as mimics of sites of virus transmission during primary HIV-1 infection [33,37]. While these models are of great value for advancing our understanding of pathological mechanisms, they do not offer tight control over critical experimental parameters such as cell density and composition, are subject to significant donor-to-donor variability, and their availability is limited. These limitations have precluded their use for defining the relative contribution of individual virus transmission modes and generated the need for the development of alternative experimental approaches. In, for example, immunology, significant efforts have been made to establish synthetic 3D systems to complement our portfolio of experimental systems with approaches that offer tight experimental control and direct access to visualization and genetic modification of individual cells. Particularly for lymphocytes and dendritic cells, 3D matrices made of collagen have emerged as the gold standard for such studies [38,39,40]. Collagen has the advantage of being a major constituent of extracellular matrix, supporting cell viability for up to several weeks, and allowing the easy removal of cells by collagenase digestion. Moreover, the good optical properties of the matrix provide access to live cell microscopy and the architecture of the matrix can be modified easily by adjusting polymerization conditions (collagen concentration, polymerization temperature, and medium used) [41,42,43]. The bovine and rat tail collagen classically used in 3D culture systems share high amino-acid similarities with the human type I collagen protein (91.1% identity for the alpha I chains and 87.4% for the alpha 2 chains), suggesting that they likely mirror the properties of human collagen. Consistent with this idea, CD4^+^ cells embedded in bovine and rat collagen matrices adopt migrating behaviors reminiscent of in vivo situations [44]. The xenogeneic effects associated with culturing human cells in this type of culture thus seem to be minimal with respect to cell migration, although species differences in molecular regulation between different collagens cannot be entirely excluded. These aspects allow quantitative insight to be gained into, for example, CD4^+^ T-cell function, such as their interaction and communication with DCs, or the ability to incorporate specific small RNAs into exosomes that match their behavior in vivo [45,46]. We therefore embarked on an attempt to reconstitute key aspects of HIV-1 spread in target tissue using 3D collagen cultures [47]. 

## 3. New Insights from Combining the Study of HIV-1 Spread in 3D Collagen Cultures with Computation 

In our recent proof-of-concept study [47], 3D cultures used to study HIV-1 spread included exclusively lymphocytes, considering HIV-1-infected and uninfected CD4^+^ T cells. Using a reporter virus expressing a sortable cell surface tag enabled the generation of pure HIV-1-infected donor cell populations that were mixed with uninfected target cells at a defined ratio prior to being embedded in 3D collagen. Cells were viable over several weeks in this system and cultures allowed the quantification of viral titers, quantification of expansion or depletion of specific cell populations, and the determination of cell motility and cell–cell contact formation/duration over time. Finally, the system allows for the use of collagen types with different density, with rat and bovine collagen resulting in denser or looser meshworks, respectively [48]. Recording the dynamics of HIV-1 spread and CD4^+^ T-cell depletion in parallel suspension and 3D cultures immediately revealed that the accurate interpretation of this plethora of quantitative kinetic data required computational approaches. Moreover, addressing the potential relationship between cell motility and virus spread, as well as disentangling the relative contribution of cell-free vs. cell-associated virus transmission to the infection dynamics, was impossible to assess experimentally but required the development of customized computational models. Our analysis relied on several iterative cycles between experimental quantification and computation and resulted in an Integrative method to Study Pathogen spread by Experiment and Computation within Tissue-like 3D cultures(INSPECT-3D) workflow depicted in Figure 2.

Together, the development and first application of INSPECT-3D revealed that HIV-1 spread is delayed relative to suspension cultures when cells are placed in a dense matrix but, after an initial phase characterized by low virus replication, follows kinetics that are similar to those seen in suspension cultures. In contrast, low-density collagen matrices did not support a marked spread of HIV-1. Several findings allowed these surprising observations to be explained. First, 3D collagen, irrespective of its density, potently suppresses the infectivity of cell-free HIV-1 particles (approx. 20-fold). Moreover, virion diffusion rates are too slow to efficiently deliver virus particles to new target cells in the spacing of 3D collagen (i.e., diffusion of virions to the nineteen nearest neighboring cells requires ~22.6 h, in comparison to half-life HIV-1 particle infectivity of t_1/2_ = 17.9 h). Together, these results revealed that 3D tissue-like environments potently restrict cell-free HIV-1 infection (see discussion of environmental restriction below). Analysis of the experimental data by mathematical modeling allowed us to estimate the relative contribution of cell-free and cell-associated HIV transmission under these various conditions. Hereby, consistent with the experimental observations, we found that HIV-1 spread in 3D collagen predominantly relies on cell–cell transmission (~78% (73−100%) and ~63% (55.8−100%) of all infections for loose and dense collagen, respectively). In contrast, in suspension cultures, there was no dominance of cell–cell transmission found and its contribution ranged widely between 0 and 100% (Table 1). Viral spread could thus be explained by the exclusive use of cell-free or cell–cell transmission. This revealed that in suspension cultures, no selection pressures for any of the transmission modes exist. This flexibility is consistent with the observation that physical separation of cells in suspension by agitation or transwells, which create conditions in which (i) cell-free infection depends on long-range diffusion of virus particles which compromises their infectivity and (ii) cell–cell contact is limited, markedly reduces virus spread [14,15]. Moreover, our analyses revealed that transmission dynamics substantially differ between suspension cultures and collagen environments. 

Since 3D collagen suppressed cell-free HIV-1 infection and virus spread depended on cell-associated virus transmission, we investigated this scenario in more detail and found that in dense 3D collagen, contacts of extended duration between donor and target cells were more frequent in comparison to in loose collagen. These findings suggested that the dense 3D environment specifically promotes cell-associated HIV-1 transmission, while the low virus replication rates observed in loose collagen stem from the combination of suppression of cell-free infectivity by the 3D matrix and the lack of support for cell-associated virus transmission due to a lack of long-lasting cell–cell contacts. Importantly, the computational model was able to predict the extent to which cell densities had to be increased experimentally to overcome these barriers to HIV-1 spread in loose 3D collagen, suggesting that adopting a 2D monolayer type of configuration could override adverse effects of a 3D matrix on HIV-1 spread. These findings also imply that in a dense matrix, in which cells are not unphysiologically densely packed, cell motility is a key parameter governing the efficacy of HIV-1 spread, i.e., allowing transport of the infection to other areas as cell-free infection is impaired. Quantitative assessments of the relationship of HIV-1 spread in specific target organs will hence require the individual reconstitution of cell and matrix density as well as of the spacing between donor and target cells.

Correlating cell–cell contact duration and productive HIV-1 transmission also allowed us to predict the minimal donor–target cell contact time required for productive infection of new target cells, which was estimated to be in the range of ~25 min. Low-resolution inspection of productive donor–target cell contacts in dense 3D collagen revealed a strikingly close and intimate organization of these contacts, and it will be interesting to dissect how this morphology translates into elevated rates of virus transmission and how dense, but not loose, 3D environments induce these events.

## 4. How Well Do Experimental and Computational INSPECT-3D Data Match In Vivo? 

Although ex vivo 3D collagen cultures allow an improved experimental assessment of physiologically relevant cellular behaviour, they still only provide a reduced representation of relevant tissue conditions and do not completely mirror specific target tissues. However, comparison of the kinetics inferred for viral transmission and cellular turnover dynamics within these culture systems can help to assess their relevance for understanding the spread of HIV-1 in vivo. Mathematical modeling has long been used as an essential tool to dissect the multifactorial complexity of viral transmission and replication dynamics (reviewed in References [49,50]). Standard models of viral dynamics describing the turnover of uninfected and infected cells and the progression of the viral load by systems of ordinary differential equations have been applied to various experimental and clinical data assessing the dynamics of HIV-1 viral spread [51,52,53]. In particular, these models allowed the quantification of rates describing the kinetics of viral production (ρ) or the death of infected cells (δ) based on patient data. Using HIV-1 plasma RNA levels as an indicator for viral spread in patients, several studies estimated an average production of ~10^10^ HIV-1 virions in total per day, with estimates of the death rate of infected cells between δ ≈ 0.46−1.40 day^−1^, corresponding to average half-lives of infected cells between t_1/2_ = 0.5 and 1.5 days (t_1/2_ = log2/ δ) [51,52,53,54]. Studies based on standard in vitro culture systems have produced estimates that are at the upper end of the half-lives determined for infected cells in vivo, with estimates of the corresponding death rates being in the order of δ = 0.5 +/− 0.1 day^−1^ [55]. The estimates on the death rate of infected cells assessed via the INSPECT-3D workflow are within the same range, with a slight tendency towards higher death rates within 3D collagen environments compared to 2D suspension cultures (δ = 0.46–0.56 day^−1^ in dense and loose collagen vs. δ = 0.40−0.44 day^−1^ in suspension; ranges based on 95% confidence intervals of estimates, Table 1). Although the differences are minor, this might indicate that environmental factors, in combination with additional clearance mechanisms such as immune responses, could also contribute to the shorter half-lives of infected cells observed in vivo. While mathematical analyses of HIV-1 viral load kinetics in patients usually allow an appropriate quantification of viral and infected cell decay dynamics, as they work under the assumption that Highly Active Antiretroviral Therapy (HAART) effectively blocks novel infections [50,52], assessing the kinetics of viral transmission and spread in vivo is much more difficult. This requires an approximation of the available number of target cells, which might vary dependent on the time-point of infection and the different compartments of viral replication [56]. In vitro systems, with their associated complete knowledge of the initial experimental conditions, provide an advantage for assessing these dynamics, but have to account for the general lack of potential physiological relevance. As cell-free and cell-associated transmission modes are synergistic, unraveling their relative contributions to viral spread in vitro usually requires the impairment of at least one of them [4,55,57,58,59]. Iwami et al. [55] set out to assess the transmission rates for HIV-1 using shaken suspension cultures where cell–cell transmission was considered to be impaired [14]. Based on their data, they estimated that cell-associated viral transmission contributed to approximately 60% of viral infections. Without the assumption of efficiently blocking one transmission mode, we found similar contributions of cell-associated transmission modes to viral spread within 3D collagen (with only ~22% (0%, 27%) and ~37% (0%, 44.2%) of infections within loose and dense collagen, respectively, due to cell-free transmission). Thus, there was a clear difference in this contribution compared to spread within 2D suspension cultures. The simultaneous analysis of all three environments revealed a ~4–7-fold higher probability of cells becoming infected by cell–cell transmission within dense and loose collagen compared to suspension. The impaired contribution of cell-free transmission within 3D collagen compared to suspension cultures is further indicated by a ~50% reduction in viral production rates, and a measured relative infectivity of virions that was reduced to ~14% within collagen (β_f_, Table 1). In addition to previous reports that blocking CD4^+^ T-cell exit from lymph nodes restricted HIV-1 spread [27], these findings obtained for viral spread within 3D collagen environments support the argument that cell–cell transmission is the main contributor to HIV-1 spread in vivo. The importance of cell-contact-dependent transmission modes in tissue-like conditions is further indicated by the increased fraction of CD4^+^ T cells predicted to be initially refractory to infection in collagen (loose ~50%; dense ~30%) compared to suspension (~17%; fractions at 2.5 days post infection). This could point towards cells that are inaccessible for contact-dependent transmission modes due to physical barriers, as they are blocked by collagen within certain areas of the culture. 

Combining live-cell microscopy data with a spatially detailed cellular Potts model in our INSPECT3D workflow also allowed us to estimate a minimal contact duration between infected and uninfected cells required for productive infection of ~25 min. Importantly, this matches very well the time required for the transfer of fluorescent viral material across the VS ex vivo [17,20], indicating that the events observed by these imaging approaches typically result in the productive infection of the target cell. 

Based on the relative comparison of the viral kinetics between 2D suspension and 3D culture systems, our study indicated the need to account for physiological conditions when aiming to quantitatively understand viral replication and transmission dynamics in vivo. Arguably, the dense collagen conditions we used [47] might come close to mucosal tissue without reflecting the cellular composition of this tissue environment. Extending the existing culture systems by incorporating additional cell types and molecular factors will be needed to approximate specific tissue conditions. Other relevant physiological parameters that will have to be considered to be implemented to advance the experimental 3D model further include dynamic tissue perfusion, which might impact HIV-1 spread, e.g., by affecting the dynamics of cell-free infections or the frequency and/or stability of cell–cell interactions. It will be interesting to assess how these additional factors might impact the quantification of the viral processes obtained so far. 

## 5. Requirements and Limitations of Current Computational Approaches to Simulating HIV-1 Spread in Tissue 

The majority of previous studies combining mathematical models and experimental data have relied on model systems based on ordinary differential equations that use time-resolved measurements of cell or viral concentrations [50]. These analyses have increased our knowledge of various aspects of HIV-1 life-cycle kinetics and transmission dynamics, and are still the method of choice for gathering most types of experimental and clinical data. However, a more detailed analysis, and thus computational representation, of single-cell dynamics within tissues is needed to understand the dynamics of HIV-1 spread within multicellular systems.

Population dynamic models have their limitations when it comes to analyzing cell-based transmission modes, as their ability to describe single-cell transmission dynamics appropriately is, by definition, hampered. Several extensions have been developed that adjust standard models for viral dynamics to account for cell-contact requirements dependent on the tissue environment [64,65]. Nevertheless, the need to clearly quantify the kinetics of cell-associated viral transmission modes, and to determine the contributions of cell-free and cell-associated transmission, as well as local immunity to viral spread, provides novel challenges for computational analysis. 

Individual-cell-based models that describe single-cell behavior have been used previously to simulate and analyze viral spread [66]. However, these model systems were mainly used for qualitative assessment of dynamics, e.g., for the spread of HIV-1 [67,68], as appropriate data and tools for model parameterization remained limited. Both limitations have now been overcome. Advanced imaging and visualization techniques used in vitro and in vivo [33,69,70] have increased our ability to observe and quantify cellular behavior and infection processes at a single-cell level. In addition, improved parameter inference tools and modeling systems have enabled us to quantitatively adapt individual-cell-based models to these novel types of data [71,72]. However, several challenges remain to appropriately analyze HIV-1 spread within different tissue environments using mathematical and computational models (Figure 3). A major challenge for computational approaches that analyze the spatiotemporal dynamics of HIV-1 spread is the appropriate representation of cell motility and the underlying tissue architecture, e.g., as represented by the collagen matrix within 3D ex vivo culture systems. Determining the influence of the environment on cell and infection dynamics requires a detailed description of these matrices and networks that structure the tissue or culture system. Previous attempts to describe the topology of the fibroblastic reticular network within lymph nodes in order to analyze T-cell motility and activation dynamics have shown the difficulty of such tasks [73,74,75,76]. High-resolution images of collagen matrices and fibroblastic reticular networks help to improve our understanding of the structures shaping and influencing tissue architecture and cell motility [48]. Nevertheless, the identification of appropriate quantities for network description, such as the pore size, fiber length, and connectivity, and how they relate to network density [42,43,77,78,79], is also required in order to provide a reliable computational representation of 3D tissue environments that accounts for the different resolutions of fibers and cells [73,76,80,81,82]. The challenges of describing cell motility based on live-cell microscopy data, and possible ways to refine these analyses, have been already discussed elsewhere [83]. Considering relevant physiological tissue conditions will also have to account for various cell types, such as T cells, dendritic cells, and macrophages, and requires a detailed characterization of their complex morphology and dynamics in 3D. In particular, this also applies to the appropriate representation of the diffusion dynamics of soluble factors, such as virions or chemokines, that are important for cell–cell communication, infection spread, or cell movement. Being able to measure gradients and concentrations of these factors, especially their release dynamics from single cells, would help to mimic their dynamics and to determine the influence of these factors on viral spread and the efficacy of the counteracting immune response [84].

Finally, a major challenge for the computational modeling of 3D tissue conditions on a single-cell level is the increased level of detail required for its description, and especially its parameterization. The higher complexity of the model systems and the adaptation of their stochastic dynamics to heterogeneous experimental data requires the use of increased computational resources, including high-performance computing clusters and sophisticated parameter inference tools. Improved parallelization and advanced parameter estimation methods, such as approximated Bayesian computing [72], could help to reduce computational run time costs. In addition, simulation areas in terms of the size and number of cells considered have to be chosen with care in order to balance the need for sufficient tissue volumes that account for stochastic variability while simultaneously ensuring computational efficiency for the simulation of long-term infection dynamics. Thus, novel advances in imaging techniques and computational methods will improve our ability to conduct data-driven computational modeling of HIV-1 spread within multicellular systems, which will be essential to disentangling and understanding the processes that shape these dynamics within physiologically relevant conditions. 

## 6. Environmental Restriction to Cell-Free HIV-1 Infection

A central finding of the studies on HIV-1 spread in 3D collagen is that residing within such a matrix reduces the infectivity of cell-free virus particles approx. 20-fold [47]. Since single-particle tracking has revealed that particles frequently undergo short and transient interactions with the collagen matrix, this negative impact on particle infectivity likely represents the consequence of physical stress imposed on virions in the context of these interactions. It is noteworthy that marked effects of the tissue environment on virus infectivity have been reported. One example is the interaction of viruses with bile acids, which can induce fusogenicity (e.g., norovirus [85]) or restrict infection (cytomegalovirus [86]; Hepatitis Delta virus [87]). Similarly, seminal fluids can impact virion infectivity via peptide-mediated enhancement of virus infectivity (e.g., retroviruses including HIV-1 [88], Ebola virus [89]) or by affecting particle aggregation [90,91]. Since the infectivity reduction in 3D collagen results from negative physical impact by the surrounding matrix and thus represents a direct effect of tissue architecture rather than the consequence of compounds released in the extracellular space, we propose the novel term “environmental restriction” for this phenomenon. Considering that this environmental restriction seems to represent a built-in antiviral activity of the 3D collagen matrix, this is conceptually similar to intrinsic immune barriers such as host cell restriction factors, and could thus be viewed as a novel element of innate immunity (Figure 4). In contrast to cell-autonomous mechanisms exerted by, for example, restriction factors, this activity would reflect a tissue-autonomous restriction. Since the restriction of virus spread by extracellular tissue architecture has only recently begun to become apparent, we can only speculate about the effector function mediating the restriction. Such environmental restrictions could include direct effects on the pathogen, such as the impairment of cell-free infectivity [92] (see below), but could also impact the host cells (e.g., via the promotion of longer-lasting cell contacts with architecture optimized for virus transmission). It can also be envisioned that, for example, transcriptional and epigenetic changes resulting from mechanosensing of the cells in the 3D environment would affect expression and activity of pro- and antiviral host cell factors. Assessing the effect of tissue-like 3D environments on the permissivity of various HIV-1 target cells for infection and their ability to sense the infection will be an important area of future research. Cell-associated HIV-1 transmission is currently considered to be beneficial for the virus, as it allows for more polarized and therefore efficient short-term transfer of particles to target cells, allows host cell restriction factors such as tetherin to be partially overcome, and avoids the recognition of certain neutralizing antibodies [25,93,94,95]. In addition, we propose that cell-associated HIV-1 transmission evolved from the need to bypass the environmental restriction against cell-free virion infectivity exerted by HIV-1 target tissue. 

Following the initial description of the environmental restrictions on cell-free HIV-1 infectivity in 3D collagen cultures, it will be important to dissect the mechanism by which tissue-like 3D matrices impair the infectivity of cell-free virions. Intuitively, physical interactions with the matrix could enhance shedding of the Env glycoprotein. However, overall Env levels in virions were unaffected by interactions of virus particles with the 3D matrix. Alternatively, the 3D matrix could affect virion infectivity by alteration of their aggregation state [90]. However, single-particle tracking of HIV-1 particles in 3D matrices did not reveal virion aggregates in either suspension or 3D collagen, suggesting that HIV-1 particles do not form aggregates under these experimental conditions. Finally, the reduced particle infectivity in 3D collagen may also reflect adaptation of the producer cells to the 3D environment; however, the reduction of virion infectivity in 3D collagen was similar when particles were produced within the matrix or embedded in a cell-free matrix, indicating that the matrix exerts direct effects on HIV-1 virions. Future studies will be thus required to dissect whether the 3D matrix affects HIV-1 infection at the level of particle entry into target cells, e.g., by affecting the conformation of Env glycoproteins on virions, or at subsequent post entry steps (Figure 5). Assessing how conserved this effect may be among viruses other than HIV-1 will be helpful in dissecting the underlying mechanisms. Another key question is whether this barrier is indeed in place in HIV-1 target tissue in infected individuals. Analyzing this question is complicated by the fact that HIV-1 particles derived from patient samples represent a pool of particles of heterogeneous and undefined origin, and that potential effects of environmental tissue restrictions could not easily be distinguished from effects due to soluble factors present in bodily fluids. Organotypic cultures will likely be instrumental for addressing this question.

## 7. Conclusions and Future Perspectives

The efficacy of virus spread in vivo is governed by a complex array of parameters that can only be disentangled by the combined application of synthetic and organoid 3D models and computational analyses. Our proof-of-concept study on HIV-1 spread in 3D collagen cultures revealed a significant impact of this tissue-like environment on modes and efficacy of HIV-1 spread, and suggests the existence of tissue-intrinsic mechanisms that suppress viral replication (environmental restriction). With the constant development of physiological ex vivo 3D culture systems and improved computational analyses, this approach is likely to continue to reveal more relevant details about the mechanisms of virus spread in vivo. An important aspect of such future efforts will be to decipher the contribution of antibody-mediated neutralization of HIV-1 during cell–cell spread (reviewed in Reference [96]) by extending the mathematical model and experimental validations within the INSPECT-3D workflow. Moreover, exosomes that resemble retroviral particles can also play a role in viral spread by affecting susceptibility to infection or the life-span of bystander cells [97,98,99]. However, the impact of tissue environments on these activities is unexplored. Finally, the INSPECT-3D workflow can directly be used to investigate the spread of other T-cell-tropic viruses, such as HTLV-1, for which the concept of cell-associated transmission via a VS was first established [10], and for which infection is thought to almost exclusively depend on cell–cell transmission, even in suspension cultures [100,101].

## Figures and Tables

**Figure 1 cells-09-01112-f001:**
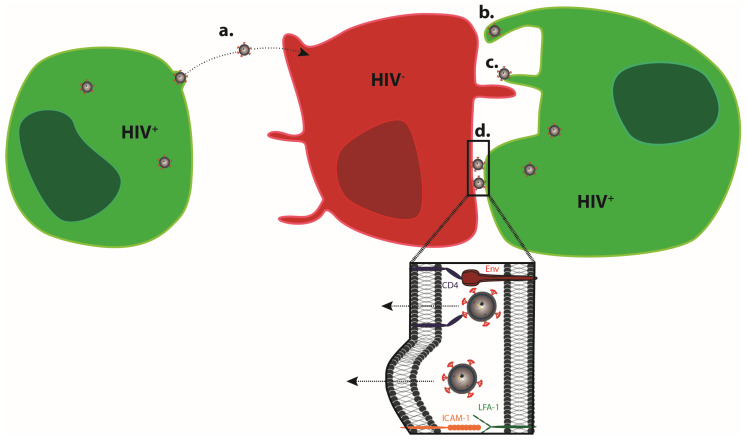
Transmission modes of HIV-1. Viral particles infect target cells via cell-free (**a**) or cell-associated (**b**-**d**) modes of transmission. (**a**) Viral particles bud at the surface of infected donor cells, mature, diffuse, and infect non-adjacent target cells. (**b**,**c**) Virions can bud at the tip (**b**) and surf along (**c**) filopodia to enter in adjacent target cells. In addition, infected and non-infected cells establish close contact, forming a virological synapse (**d**). Whether HIV-1 enters the target cell via fusion at the plasma membrane or following prior internalization [30,31] remains a matter of debate, and may depend on the nature of the target cell (reviewed in Reference [32]).

**Figure 2 cells-09-01112-f002:**
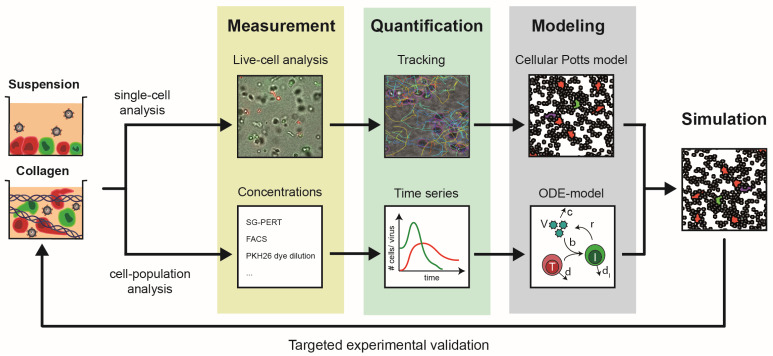
Schematic of the INSPECT-3D workflow that combined single-cell and cell-population measurements by mathematical modeling to reveal the dynamics of infection spread. Infected and non-infected Peripheral Blood Mononuclear Cells (PBMCs) were cultured in suspension or collagen cultures at a 1:20 ratio. In suspension cultures, cells rapidly settled at the bottom of the culture dish to form dense 2D monolayers. In collagen, cells were spatially separated from another, migrated along the 3D scaffold, and eventually made contact with other cells. Measurements from live-cell imaging of single cells and bulk dynamics allowed quantification of individual cell motilities and interactions, as well as long-term kinetics of viral load and cell concentrations. Appropriate mathematical model systems that either rely on spatially resolved cellular Potts models describing single-cell behavior or systems of ordinary differential equations (ODE) enabled us to disentangle processes of cell motility, cellular turnover, and viral transmission and replication. Theses analyses led to an integrative computational model for HIV-1 spread that provides a mechanistic and quantitative understanding of viral transmission within these multicellular systems. Recurrent experimental validation of model predictions by in silico manipulation of experimental conditions was used to refine the model system and reveal key processes governing the dynamics of HIV-1 spread.

**Figure 3 cells-09-01112-f003:**
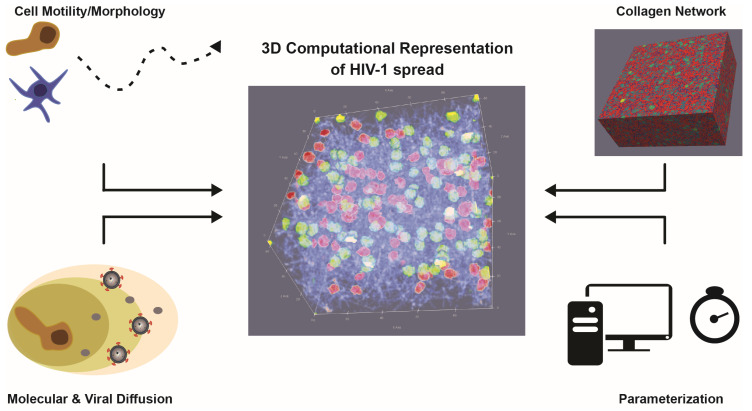
3D computational representation of HIV-1 spread: Appropriate representations to infer the processes that govern the dynamics of infection require a detailed description and quantification of individual cell motility and morphology, the underlying tissue structure, and the diffusion dynamics of soluble components (chemokines, virions, etc.) within these environments. Data-driven parameterization and subsequent simulation of these 3D model systems requires computationally efficient methods. The sketch shows a computational representation of 100 HIV-1-infected (green) and 100 uninfected (red) cells within a 3D collagen matrix (blue fibers).

**Figure 4 cells-09-01112-f004:**
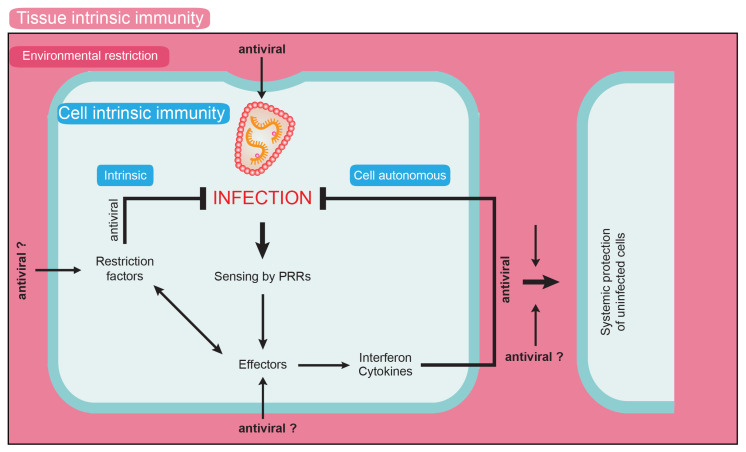
Schematic representation of the environmental restriction concept. Human cells bear cell-intrinsic mechanisms to limit virus spread, such as intrinsic immunity by restriction factors and cell-autonomous immunity by recognition of viral components by pattern-recognition receptors (PRRs). Both pathways can lead to the induction of antiviral effectors such as interferons and cytokines, which can impair virus replication in both the infected and bystander cells. In addition, we propose that the tissue environment can provide tissue-intrinsic immunity by exerting environmental restrictions on virus replication. Such activities could be exerted via direct effects on virus particles, but we also hypothesize that tissue interactions impact the expression and activity of cell-autonomous immune mechanisms.

**Figure 5 cells-09-01112-f005:**
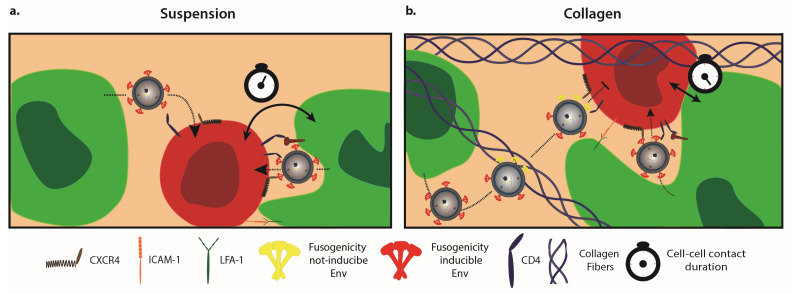
HIV-1 spread in suspension and 3D collagen cultures. (**a**) In suspension cultures, virions can be transmitted to uninfected target cells via both cell-free and cell-associated modes of transmission. (**b**) In collagen cultures, cell-associated transmission is the main driving force for HIV-1 spread. This reflects on one hand that the infectivity of cell-free particles is impaired, which coincides with frequent transient interactions with collagen fibres. These interactions could impair the ability of Env proteins to adopt a fusogenic conformation in response to interaction with the HIV-1 receptor/coreceptor complex on target cells, and thus inhibit virus entry. Moreover, cell-associated HIV-1 transmission may benefit from alterations in the architecture of contacts between donor and target cells, which are more stable and involve larger areas of membrane compared to cells in suspension.

**Table 1 cells-09-01112-t001:** Estimates for HIV-1 infection dynamics obtained from the analysis of experimental infection in cell lines or primary cells and in vivo patient data. Obtained estimates vary over several orders of magnitude due to different mathematical models and data types used. Please note that viral production rates vary in terms of units due to different quantification methods used. Numbers marked with (*) indicate estimates obtained for Simian Immunodeficiency Virus (SIV) infection.

Parameter	Description	Unit	Cell Lines	Primary Cells [47]	In Vivo
2D Suspension	3D Collagen
Loose	Dense
ρ	Viral production rate	×10^4^ day^−1^	2.61(1.55, 3.70)(RNA copies) [60]	1.02 (0.80, 1.38)(RT cell^−1^)	0.48 (0.37, 0.66)(RT cell^−1^)	0.43 (0.32, 0.61)(RT cell^−1^)	5.0 * (1.3, 12.0)(virion cell^−1^) [61]
0.07–0.34 [56,62]
ρI	Total viral production rate	×10^10^ virions day^−1^					1.03 ± 1.17 [52,53]
δ_I_	Death rate of infected cells	day^−1^	0.50 ± 0.10 [55]	0.42 (0.40, 0.44)	0.48 (0.46, 0.50)	0.52 (0.51, 0.56)	0.48–1.36 ± 0.16 [53,63]
0.7 ± 0.25 [51]
1.0 ± 0.3 [54]
0.88 * ± 0.40 [61]
β_f_	Cell-free infection rate	×10^−5^ day^−1^	0.42 ± 0.14 [55](p24)	2.3 (0, 3.0)(RT)	0.14 × suspension	0.14 × suspension	
β_c_	Cell–cell infection rate	×10^−5^ (cell × day)^−1^	0.11 ± 0.03 [55]	10^−6^ (0, 7.0)	4.3 (3.6, 5.4)	1.7 (1.4, 2.6)	
P_f_	Predicted percentage of infections by cell-free transmission	%	57% ± 7% [55]	99.6% (0.0%, 100%)	22.0% (0.0%, 27.0%)	37% (0.0%, 44.2%)

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
