# Peer review of "Environmental Restrictions: A New Concept Governing HIV-1 Spread Emerging from Integrated Experimental-Computational Analysis of Tissue-Like 3D Cultures"

_cells, 2020, doi:10.3390/cells9051112_

Round 1
Reviewer 1 Report
Ahmed et al. manuscript reviews the main findings originated from the recently published paper: Imle et al. Nature Communications 2019, and expand them integrating with current reports regarding different physiological systems. The purpose of the review is to improve the INSPECT-3D system and make it applicable to a wider range of research models that mimic diverse physiological conditions. The authors first list the limitations of 2D cultures, being the most common method widely used to mimic in vivo systems, and of ex vivo samples, as attempt to reconstitute an organ-like tissue (organotypic cell system). Then, the authors enforced the idea of using collagen-based 3D cultures combined with computational models (INSPECT-3D), in order to expand the knowledge about the relative contribution of cell-free vs cell-associated virus transmission. They emphasize the ability of dense vs loose 3D-collagen systems to distinguish and quantify the contribution of cell-to-cell transmission over cell free transmission, which is not supported in those systems. They also propose to expand the existing culture systems by mimicking additional physiological conditions, including cells of diverse nature or molecular factors in the system. Finally, the authors discuss some of the environmental restrictions that may impair the cell-free transmission of HIV in vivo and in 3D-collagen models.
The manuscript deals with a crucially significant and interesting topic, important for the enhancement of cell culture conditions and development of models to study viral transmission. Importantly, the insights provided in this review can be applied to studies regarding different viruses, not only HIV-1. Overall, the paper represents an efficient plan of action for the development and expansion of the INSPECT-3D collagen-based system to model and analyze diverse scenarios of HIV-1 infection spread in vivo.
Overall, the review is well written.
Introduction and frame of the problem are well presented, and the manuscript is written in a correct English language and style.
Comments:
- Although the review is well written and well developed, the importance of the role of environmental restrictions is not well emphasized. The title highlights the concept of environmental restrictions as being the main topic of the review. This leads the reader to expect a wider discussion about those factors. Instead, the authors provide a very compelling and complete introduction to the problem, a very detailed description of the 3D collagen systems and their advantages/limitations, and they limit the discussion about environmental restrictions as last. My suggestion is to change the title, framing more precisely the topic of discussion of the review, or to expand the environmental restriction session, to be the main topic.
- Line 159: as well AS long term kinetics. “As” of as well as is missing
- Line 185-188: Were the infected donors and targets separated by trans-wells in order to estimate the real contribution of cell-to-cell transmission in the suspension culture system? This result is in contrast with few reports, such as Orlandi et al. JIM, 2016, where the authors showed that the cell-to-cell transmission of HIV-1 in 2D cultures can be highly decreased limiting the physical contact between infected donors and uninfected targets by the use of trans-wells.
- Lines 225-228 and Table 1: Can the authors better define the correlation between death of infected cells and half-life of infected cells? In different points of the main text data is expressed as “half life” of infected cells and in some others, including Table 1, data is expressed as death rate. Please explain the difference or alternatively, authors may choose to be consistent.
- Table 1: under 3D collagen title, is that “suspension” supposed to be dense?
- Table 1: under Ex vivo title, is that “suspension” supposed to be 2D?
- Line 256 and Table 1: Can the authors better define the differences between 2D suspension ex vivo and in vitro?
- It would be critical to stress more about neutralizing and non-neutralizing antibodies in the context of the 3D-collagen based system among the environmental restrictions. Adapting and/or developing the INSPECT-3D mathematical model with the purpose of defining the role of mAbs in the cell-to-cell transmission mode of HIV-1 would be of critical interest. Also, how would it be the contribution of mAbs with dense vs loose collagen? How do the author predict to expand the mathematical models to take in account the additional parameter of having antibodies in the system? The authors may include this point in the review.
Reviewer 2 Report
This is an interesting review discussing the difficulties in estimating cell-free and cell-to-cell HIV and other retroviral spread in cell suspension cultures, 3D matrix cultures and mouse models. The simplest analysis can be done in a dense suspension cells. Adding collagen matrices or using explants or in vivo models substantially increases the complexity of HIV spread analysis. In such experimental conditions there are limits in accurate measure of cell survival, mobility, infection e.t.c., and different computational approaches can help to shape a whole picture of infectious process by operating with multiple experimentally driven parameters. Finally, the authors propose a new model of so called “environmental restriction” imposed by matrix collage fibers within which infected, uninfected cells and virus are dynamically exist, and which spatially restrict cell-free infection, impact cell-cell contact duration and so on. The model was build based on research paper published by authors earlier in Nat Communic 2019. This review is well structured and well written, will be interesting for HIV biologists, thought raises some questions that need to be addressed.
- While the molecular mechanisms of cellular restriction are quite established, the mechanisms and molecules involved in the “environmental restriction” are not known. Thus, it will be premature to refer this novel concept as an “element of innate immunity” (line 385 and fig.4), as well as to say “built-in antiviral activity of the 3D collagen” (line 383).
- In Nat Commun paper, low and high dense collagen monomers (rat and bovine, respectively) have been used to build 3D matrix. This is raising an important question how xenogeneic materials impact cells and virus attachment, motility e.t.c. This was not discussed anywhere, and should be added.
- Dissecting cell-to-cell and cell-free components of HIV transmission is very difficult task even in cell suspension and depends on a method used. Derse and Mazurov (Plos Path 2010, JVI 2015) developed an elegant intron-containing reporter system that allows measuring infection in cell co-cultures. Notably, unlike other reports but similar to author’s Nat Communic paper (lines 250-256), they demonstrated that HIV-1 spreads only twice more efficiently via cell-cell contact than via cell-free viral particles, but for HTLV-1 this difference was ten thousand. There may be a possibility that 3D matrix poses a physical barrier to cell-free infection that facilitate dissecting two modes of HIV transmission, and it will be interesting to test 3D tissue environment and INSPECT-3D for HTLV-1. This can be discussed in review.
- If we compare 2D cell suspensions, 3D tissue, and mouse model, the complexity of parameter measurement is increasing, while the possibility of experimental modulation and accuracy of measurement is decreasing. However, an important factor related to a tissue perfusion occurred in vivo relative to static fluid condition in 3D tissues were omitted in this review. Meantime, this can contribute to a cell-free infection delay and promotion of cell-cell HIV transmission in 3D cultures. This should be discussed in order to refine the side-by-side comparison of different models of HIV spread.
Minor points.
- Line 45. In ref.5 there is no direct evidence for HSV cell-to-cell transmission.
- Lines 52-53. “The patho-physiological relevance of cell-associated transmission”. Interestingly, non-infectious transmission of exosomes (J Am Soc Nephrol. 2011 Mar; 22(3): 496–507) to non-permissive cells can cause some pathology.
- Line 86. References should be added, i.e. Cell Host Microbe. 2011 Dec 15;10(6):551-62. doi: 10.1016/j.chom.2011.10.015, Cell. 2009 May 1; 137(3): 433–444.
- Line 250. Confusing references 14 and 50, make it clear.
- Line 398. Reference Viruses. 2019 Apr 26;11(5). pii: E390. doi: 10.3390/v11050390 should be added.
- In the manuscript text there is no reference to the Fig.1.
